



# Quantifying the influence of coastal flood hazards on building habitability following Hurricane Irma

Benjamin Nelson-Mercer[1], Tessa Swanson[2], Seth Guikema[1,2], Jeremy Bricker[1,3]

[1]Civil and Environmental Engineering, University of Michigan, Ann Arbor, Michigan, USA
[2]Industrial and Operations Engineering, University of Michigan, Ann Arbor, Michigan, USA
[3]Civil Engineering and Geosciences, Delft University of Technology, Delft, the Netherlands

*Correspondence to*: Benjamin Nelson-Mercer (bnelsonm@umich.edu); Jeremy Bricker (jeremydb@umich.edu)

**Abstract.** Appropriate management of coastal flood risk is critical for creating resilient communities. An important part of this is estimating what buildings will become uninhabitable due to a flood event such as a tropical cyclone. To increase the
accuracy of these estimations, habitability functions are developed to quantify the relationship between hydrodynamic hazards and the probability of a building becoming uninhabitable following Hurricane Irma. Hazards like maximum flood depths are determined by modeling Hurricane Irma flooding in Delft3D-FM coupled with the wave model SWAN. These modeled hazard levels are then extracted at building locations where Location Based Services (LBS) data provide information on buildings that were uninhabitable following Hurricane Irma. The developed habitability functions provide valuable insights into how
different hydrodynamic parameters and regression models perform for estimating building habitability, where maximum depth is generally the best predictor of habitability. Furthermore, we find that while wooden structure habitability is significantly influenced by hazard level, concrete structure habitability is not. These findings provide novel methods for estimating coastal flooding induced building uninhabitability, enhancing how planners can prepare for floods.

## 1 Introduction

Coastal flooding induced by tropical cyclones is a significant driver of structural damage, economic loss, and both short-term and long-term migration worldwide. Sea level rise and precipitation intensification resulting from climate change is expected to exacerbate the damage and loss caused by tropical cyclones (Gori et al., 2022; Hughes & Zhang, 2023; Mendelsohn et al., 2012; Woodruff et al., 2013). The number of people living in low-elevation coastal zones is also increasing, with over a billion people expected to be living in these zones by 2060 (Neumann et al., 2015). In the United States, tropical cyclones (or
hurricanes) make up the majority of costs due to all billion-dollar natural hazards, resulting in almost 7 thousand deaths and over $1.4 trillion in costs (CPI-Adjusted) since 1980 (Smith, 2020). The significant loss due to tropical cyclones and increased risk posed by climate change highlights the need for improved planning and adaptation for coastal areas subject to hurricanes.



Common tools for managing flood risk include using "damage functions" or "fragility functions" to estimate and predict the structural damage sustained during a flood event (Diaz Loaiza et al., 2022; Pistrika & Jonkman, 2010; Suppasri et al., 2013;
Tomiczek et al., 2013; Tsubaki et al., 2016; Xu et al., 2023). Typically, damage functions estimate the percent of a building damaged, while fragility functions estimate the likelihood of a building reaching a specific damaged state. These functions most commonly estimate structural damage as a function of flood depth; however, other hydrodynamic parameters such as flow velocity, unit discharge, and flood duration have also been used to estimate damage due to coastal flooding (Charvet et al., 2015; De Risi et al., 2017; Diaz Loaiza et al., 2022; Nofal et al., 2020; Xu et al., 2023). Many of these functions also
incorporate structural components to increase the accuracy of predicting physical damage to buildings (Charvet et al., 2015; De Risi et al., 2017; Paprotny et al., 2021; Tomiczek et al., 2013; Xu et al., 2023).

While damage functions are helpful for predicting structural damage, they are generally applied to derive economic losses following a flood event (Pistrika & Jonkman, 2010). Paul et al. (2024) point out the use of post-disaster economic loss to characterize risk often incorrectly emphasizes wealthier people as being at greater risk from disasters, when previous studies
have shown lower income groups are impacted more by natural disasters (Fothergill & Peek, 2004; Hallegatte et al., 2020). Fragility functions offer an improvement over damage functions in this context by predicting what state a building is in following an event such as "no damage", "moderate damage" or "complete damage" (Charvet et al., 2015; De Risi et al., 2017), but these functions are still focused only on structural damage. Assessing building habitability rather than building damage following an event is one option for providing a more equitable overview of coastal flood risk and post-disaster recovery (Paul
et al., 2024). Different factors such as structural components (number of stories, building material, etc.), power outages, school closures, socioeconomic statuses, and access to other essential services can influence if and when a building becomes habitable (Loos et al., 2023; Paprotny et al., 2021; Paul et al., 2024; Suppasri et al., 2013; Thieken et al., 2005; Yabe et al., 2020). However, physical damage to structures is often the largest factor determining a building's habitability (Paul et al., 2024), showing the importance of flood hazard consideration in predicting post-disaster building habitability.

Efforts have been made to quantify the influence of physical damages on post-disaster recovery (FEMA, 2024a, 2024b; Nofal et al., 2024; Yabe et al., 2020). Yabe et al. (2020) utilized mobile phone data to estimate immediate and long-term household displacement from Hurricane Irma, finding that housing damage rates were strong estimators of household displacement 0 days after Irma and housing damage rates were only weakly correlated with displacement 160 days after Irma. This study relied on the Federal Emergency Management Agency's (FEMA) Individuals and Households Program for estimating housing
damage, neglecting the actual flood hazard (Yabe et al., 2020). Nofal et al. (2024) transformed building fragility curves to functional fragility curves by estimating conditional probabilities of functionality states given different damage states. While habitability is considered a part of the functionality estimated by these curves, the conditional probabilities used are derived from the authors' judgement and are not directly developed from flood depths (Nofal et al., 2024). Hazus, a tool developed by





FEMA, is capable of estimating building habitability with hazard information (FEMA, 2024a, 2024b). The Hazus Earthquake
Model estimates building habitability with both demographic data and computed structural damage derived from earthquake
hazard information (FEMA, 2024a). While the Hazus Flood Model also incorporates demographic data for estimating
habitability, the hazard information used is simply the area of a census tract with nonzero inundation (FEMA, 2024b). This
exhibits a significant knowledge gap in how varying levels of flood hazards influence building habitability.

To improve coastal communities' resilience to hurricanes, this study aims to uncover the relationship between flood hazards
and building habitability following Hurricane Irma. Hurricane Irma made landfall in September 2017 in the Florida Keys as a
Category 4 hurricane before reaching southern Florida as a Category 3 hurricane (Cangialosi et al., 2021), resulting in
approximately $64 billion in damages (CPI-Adjusted) (Smith, 2020). Irma caused widespread destruction through storm surge,
wind, and wave damage, which displaced millions of people (Issa et al., 2018; Joyce et al., 2019). Through Location Based
Services (LBS) data collected from cell phones, we know if and when many buildings were once again occupied following
Hurricane Irma (Swanson & Guikema, 2024). Combining this LBS dataset with an integrated hydrodynamic-wave model of
Hurricane Irma, we draw upon previous methods for developing damage and fragility functions and apply them to develop
habitability functions. These habitability functions offer new estimates of the probability of buildings being uninhabitable
following tropical cyclones, advancing current approaches to quantifying flood-induced building uninhabitability.

## 2 Data and methods

### 2.1 Flood model development for Hurricane Irma

Coastal flooding caused by Hurricane Irma is modeled with D-Flow Flexible Mesh (D-Flow FM) coupled with SWAN
(Simulating WAves Nearshore). Hydrodynamics are simulated by D-Flow FM, which implements a finite volume solver to
calculate unsteady flow with the non-linear shallow water equations to simulate storm tide resulting from tidal and
meteorological forcings (Deltares, 2022a). The depth-averaged approach is used for this study. SWAN is a phase-averaged
wave model that simulates wave evolution (Deltares, 2022b). These models are integrated together in the Delft3D Flexible
Mesh modeling suite via online coupling, enabling hydrodynamic parameters from D-Flow FM and wave parameters from
SWAN to be exchanged every coupling timestep.

The model developed for this study includes both Collier and Monroe Counties. The extent the model is from 22.74° N to
30.94° N and 78.91° W to 84.21° W, covering the majority of Florida (Fig. 1a). D-Flow FM enables the use of an unstructured
mesh for simulations. The unstructured mesh created for this modeling has a coarse resolution of 10 km and is refined to 80 m
in areas with both coastal flooding during Irma's landfall and LBS data (Fig. 1b&c). For wave modeling, SWAN requires



nested structured meshes. Our SWAN models each have a coarse 10 km resolution mesh spanning the entire domain with nested meshes down to a refinement of 150 m for the same areas refined in the D-Flow FM model.

Digital elevation models (DEMs) used for this flood modeling come from NOAA's National Centers for Environmental
Information's (NCEI) DEM Global Mosaic and the General Bathymetric Chart of the Oceans (GEBCO). The refined areas of the flood model utilize 3 and 1 arcsecond DEMs from the NCEI's DEM Global Mosaic (NOAA NCEI, 2022). The coarser portions in the model use GEBCO's 15 arcsecond dataset (GEBCO, 2023).

Spatially varying Manning's coefficients of roughness are used to account for bed friction in the model. These values are derived from the 2019 National Land Cover Database (NLCD) for the Contiguous United States (Dewitz & USGS, 2024).
These NLCD land cover values are then converted to Manning's roughness coefficients by taking the corresponding minimum Manning's value listed in the HEC-RAS 2D User's Manual (Hydrologic Engineering Center, 2021).

Meteorological forcings used for the flood model are wind and atmospheric pressure fields. These fields are generated with the Holland model (Holland, 2008; Holland et al., 2010), which requires information on a tropical cyclone's path such as the coordinates of the eye's path, maximum wind speeds, and radius of maximum winds. The necessary Hurricane Irma best track
data comes from the National Hurricane Center's revised Atlantic hurricane database (HURDAT2) (Landsea & Franklin, 2013), supplemented by the Tropical Cyclone Extended Best Track Dataset (EBTRK) that provides radius to maximum winds information (Demuth et al., 2006). Together, these datasets and the Holland model are used to develop a symmetric profile of Irma as a spiderweb grid, which conveys the atmospheric pressures, wind velocity magnitudes, and wind directions used in the flood models on a polar grid (Deltares, 2022a). A second Irma profile is also created to account for asymmetries in the
hurricane profile. This was done by incorporating a dependency on the azimuthal angle into the Holland model used (Xie et al., 2006), enabling an asymmetric Irma profile to be generated.

The default wind drag coefficient formulation in D-Flow FM is utilized for determining the shear stress on the flow due to wind forcings. This drag coefficient is based on the Smith and Banke (1975) relationship, where the drag coefficient varies linearly from 0.00063 to 0.00723 for wind speeds from 0 to 100 m/s. It was determined that the default SWAN drag coefficient
profile was insufficient for this modeling, which relies on the Wu (1982) relationship, and an increased drag coefficient profile is needed. For SWAN, the increased drag coefficient relationship used is as follows:

$$C_D = \begin{cases} 0.0022, & U_{10} < 7.5 \\ 0.000127 U_{10} + 0.00125, & U_{10} \geq 7.5 \end{cases} \tag{1}$$



where $C_D$ is the drag coefficient and $U_{10}$ is the wind speed 10 m above the surface in m/s (Deltares, 2022b; Wu, 1982). Implementing this increased drag profile was done by increasing the wind field values by 25%, which is the wind speed
corresponding to the same wind wave growth due to the increased drag profile described by Eq. (1).

Tidal boundary conditions for the Atlantic Ocean and Gulf of Mexico are located around the northern, eastern, and western boundaries of the domain where the bed elevation is below mean sea level (MSL). Tidal constituents at these boundaries are generated from the Oregon State University Tidal Inversion Software (Egbert & Erofeeva, 2002), which are then used as astronomical forcings at the boundaries.

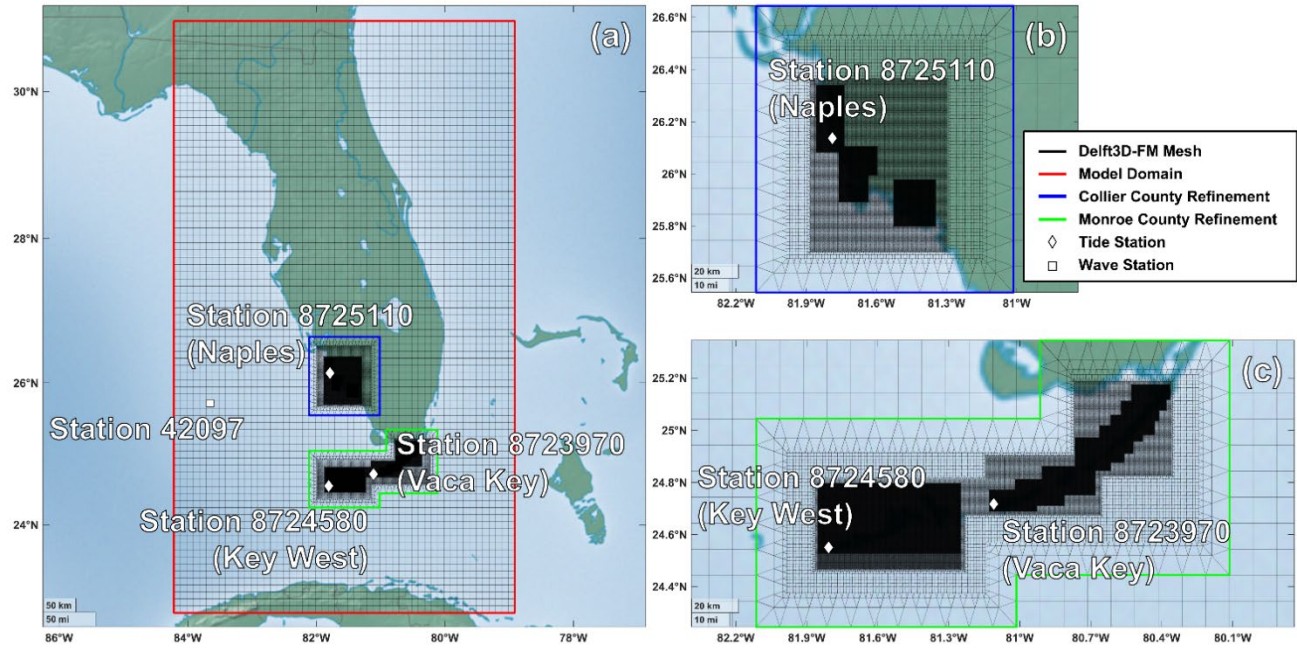


**Figure 1: Overview of the entire model domain (a) and two locations of refinement for Collier (b) and Monroe (c) Counties. NOAA tide and wave stations are indicated with diamonds and squares, respectively.**

### 2.2 Model validation

The validity of the model is assessed using water level measurement from three Florida locations: Naples, Key West, and Vaca
Key, corresponding to NOAA tide stations 8725110, 8724580, and 8723970, respectively (Fig. 1). Additionally, modeled wave parameters are compared to significant wave heights and peak wave periods measured at NOAA station 42097. First the tidal boundary conditions are validated by comparing the modeled water levels without meteorological forcings against the predicted water levels. Then the developed Irma wind and pressure fields are implemented into the model and the resulting water levels and wave parameters are validated against observations (Fig. 2). Four combinations of the symmetric and





asymmetric Irma profiles are compared: the symmetric profile is used for both D-Flow FM and SWAN (M1), the asymmetric profile is used for both D-Flow FM and SWAN (M2), the symmetric profile is used for D-Flow FM and the asymmetric profile is used for SWAN (M3), and the asymmetric profile is used for D-Flow FM and the symmetric profile is used for SWAN (M4). The root mean square error (RMSE) between modeled and observed water levels and wave parameters is determined for each model at each of the NOAA stations shown in Fig. 1 (Table 1). To remain consistent with the 30-minute time resolution of the model output, RMSE is calculated using observed data for each half hour. The difference between maximum modeled and maximum observed water levels and wave parameters is also determined at each station (Table 1).

Comparison of the four different models clearly shows the symmetric Irma profile performs the best for modeling wave parameters, where the two models that utilize a symmetric profile for SWAN (M1 and M4) have the lowest RMSE and differences in maximum modeled and maximum observed significant wave height and peak wave period (Table 1). Assessing the results of the water level validation is not as straightforward. At the Naples station, M1 has the strongest agreement between maximum modeled and observed water levels but the worst RMSE, while M4 has the worst agreement between maximum modeled and observed water levels but the best RMSE. At the Key West and Vaca Key stations, M2 performs the best for both metrics analyzed.

Two models are selected for developing habitability functions based on these performance metrics. The M1 model is used for Collier County and the M4 model is used for Monroe County. The M2 and M3 models are not considered for developing the habitability functions because the symmetric Irma profile performed significantly better than the asymmetric profile for modeling wave parameters in SWAN. Since the habitability functions are developed using maximum values of the model output, M1 is selected for Collier County to minimize the difference between the maximum modeled and maximum observed water levels at the Naples station. Between M1 and M4, the M4 model performed better for the Key West and Vaca Key stations, which is why the M4 model is used for developing habitability functions for Monroe County.

**Table 1: Goodness of fit for different combinations of symmetric and asymmetric Irma wind profiles.**

| Station | RMSE | | | | Max Modeled – Max Observed | | | |
|---|---|---|---|---|---|---|---|---|
| | M1 | M2 | M3 | M4 | M1 | M2 | M3 | M4 |
| 8725110 (Naples) | 0.6543 m | 0.5182 m | 0.6368 m | 0.5170 m | 0.0184 m | 0.8784 m | 0.1197 m | 0.8812 m |
| 8724580 (Key West) | 0.3319 m | 0.2755 m | 0.3265 m | 0.2766 m | -0.5331 m | -0.2630 m | -0.4638 m | -0.3311 m |
| 8723970 (Vaca Key) | 0.3742 m | 0.3390 m | 0.3782 m | 0.3480 m | 0.0994 m | -0.0588 m | 0.0824 m | -0.0919 m |
| 42097 (Sig. Wave Height) | 1.1357 m | 1.3990 m | 1.4063 m | 1.1394 m | 0.2840 m | -1.1610 m | -1.2400 m | 0.3270 m |
| 42097 (Peak Wave Period) | 2.1597 s | 3.0054 s | 3.0166 s | 2.1448 s | -1.6600 s | -3.5120 s | -3.5120 s | -1.6600 s |



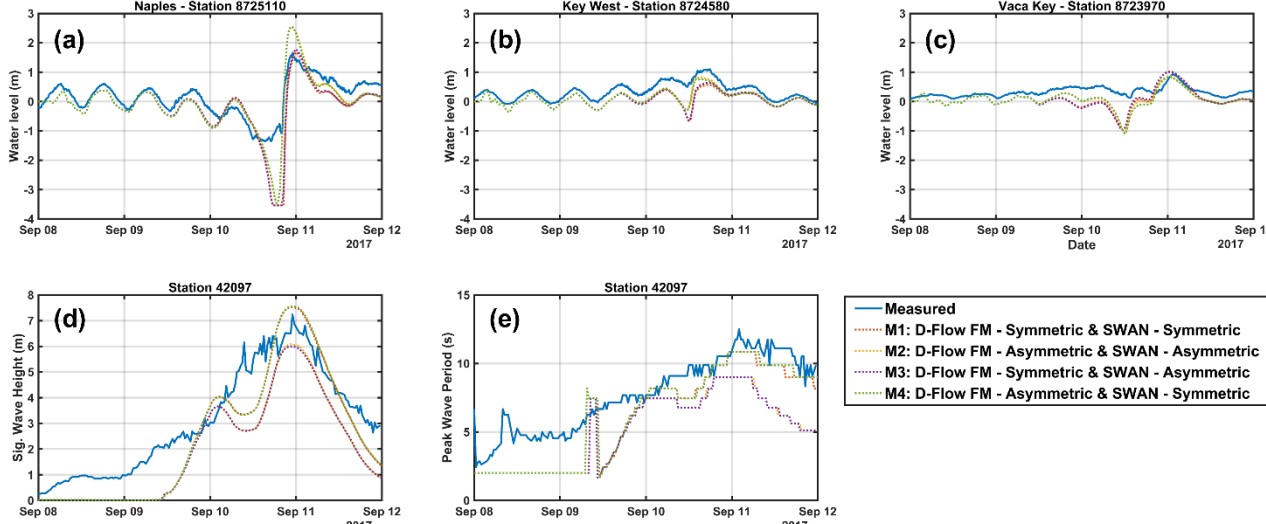

**Figure 2: Storm tide comparisons between the measured and modeled water levels relative to MSL (a-c). Comparisons between the measured and modeled significant wave heights (d) and peak wave periods (e).**

### 2.3 Determining building habitability following Irma

Whether or not a building was habitable directly following Hurricane Irma is determined using Location Based Services (LBS) and CoreLogic property data. LBS data is provided by Veraset, LLC and consists of "pings" that represent exchanges between mobile phones and a cellular network or Wi-fi. Each ping includes an anonymized user identification number, latitude, longitude, and timestamp as well as estimates of horizontal accuracy and device type. Pings are filtered and aggregated based on frequently visited locations and time of day to identify each user's home and workplace (Swanson, 2023; Washington et al., 2024). In total, there are 18,505 users with identified home and work locations available for Collier and Monroe Counties, where 16,769 of these are for Collier County and 1,736 are for Monroe County.

The recovery period for each user following Hurricane Irma is determined using a Bayesian belief network (BBN) in combination with anomaly detection methods (Swanson, 2023). The BBN incorporates contextual knowledge and time-series data of each user's daily location visits to estimate the joint probability of a user's presence at home or work on a given day prior to Hurricane Irma's landfall. By considering dependencies—such as the day of the week, prior appearances, and visits to other locations on the same day—the model identifies probabilistic patterns for all Florida users and refines these priors with individual user data to create personalized models of each user's "typical" behavior. Anomaly detection methods are applied to user data during the period surrounding Hurricane Irma's landfall to identify anomalous patterns of behavior, such as being absent from home or work or exclusively staying at home, that differ from their previously typical appearance behavior.



Recovery is defined as the date when a user's anomalous behavior ends and their visit patterns resemble their pre-landfall behavior for at least three consecutive days. Greater details on identifying recovery periods from LBS data is available in Swanson (2023). Locations where users did not recover their previous visit patterns by the end of September 28, 2017 are

assumed to be uninhabitable due to damages caused by Irma since essential services such as power and schools were recovered by this point (Hodge & Lee, 2017; Swanson, 2023). About 13.5% of the users in Monroe County and 6.0% of the users in Collier County are identified as having uninhabitable homes by this date.

Each location derived from the LBS data is then approximated to the nearest building by assigning it to the nearest CoreLogic coordinate, representing the center point of a property. This ensures each location corresponds to an actual building and

provides information on the building material. Buildings with multiple LBS datapoints assigned to it are assumed habitable if at least one LBS user returned by the end of September 2017 and uninhabitable if all users assigned to the building did not return by the end of September 2017. LBS datapoints farther than 0.001 decimal degrees from the nearest CoreLogic coordinate are excluded.

## 3 Results

### 3.1 Developing habitability functions

For each CoreLogic property location that has a habitable or uninhabitable designation from the LBS data analysis, the maximum depth, velocity, and significant wave height experienced are determined by matching each building's latitude and longitude to the nearest cell in the computational mesh of the flood model (Figs. 3 and A1-A2). If a building's coordinate is inundated at the initialization of the model, indicating its corresponding mesh cell's bed level is below mean sea level, the

building is excluded from our analysis. Additionally, buildings with a maximum depth of zero, determined from the hydrodynamic model, are removed. After these exclusions, there are 920 locations with assigned hydrodynamic parameters, where 350 of these locations are for Collier County and the other 747 locations are for Monroe County.

From the 920 locations included in our analysis, 110 of these buildings do not have the user returning by the end of September 2017, indicating these 110 buildings were uninhabitable due to Hurricane Irma. 84 of these uninhabitable buildings are in

Monroe County and the other 26 are in Collier County.



**Figure 3: Maximum modeled flood depths for Collier County (a) and the western (b) and eastern (c) regions of Monroe County. Building locations and associated maximum flood depths used for habitability functions (d-f). To preserve privacy the exact building locations are not identified.**





These outputs are used to develop habitability functions for Florida due to Hurricane Irma as a function of the modeled maximum depth, flow speed, and significant wave height (Fig. 4). Since each datapoint's habitability entry is binary (habitable/uninhabitable), logistic regression is used to develop habitability functions.

$$P(y = 1) = \frac{1}{1 + e^{-(\beta_0 + \beta_1 X)}} \tag{2}$$

where $P(y = 1)$ is the probability of a building being uninhabitable, $X$ is the hydrodynamic hazard level, and $\beta_0$ and $\beta_1$ are

the logistic regression coefficients. Maximum likelihood estimation is used to estimate the values of the coefficients. Additionally, the 95% confidence interval is determined to assess the uncertainty of each function (Fig. 4). Goodness of fit for the developed habitability functions is determined with the Akaike information criterion (AIC) and Bayesian information criterion (BIC) (Akaike, 1974; Schwarz, 1978), where lower values of AIC and BIC indicate a better fit.

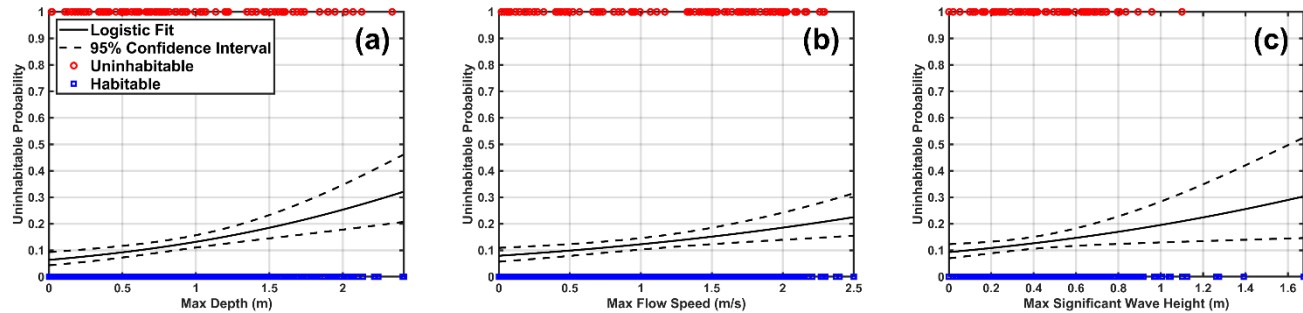

**Figure 4: Building habitability as a function of maximum depth (a), flow speed (b), and significant wave height (c).**

**Table 2: Coefficients for maximum depth, flow speed, and significant wave height.**

|  | Depth | Flow Speed | Sig. Wave Height |
|---|---|---|---|
| $\beta_0$ | -2.688** | -2.454** | -2.278** |
| $\beta_1$ | 0.803** | 0.486** | 0.864* |
| AIC | 660.521 | 666.073 | 671.668 |
| BIC | 670.170 | 675.722 | 681.317 |
| $\chi^2$ test p-value | 3.689e-05 | 7.060e-04 | 0.0153 |

For individual coefficients: * p-value < 0.05, ** p-value < 0.001

All three habitability functions developed show positive relationships between hazard level and uninhabitable probability that are significant at the 95% confidence level (Table 2). This indicates that buildings that experienced larger flood depths, flow





speeds, and wave heights were more likely to be uninhabitable following Hurricane Irma. Of the three habitability functions
developed, the one dependent on depth performs the best, having the lowest AIC and BIC values. Conversely, using significant
wave height to predict building uninhabitability shows the worst fit.

**3.2 Influence of building material on habitability**

The exterior wall material listed for each building is the building material information available for locations in Monroe
County. Collier County does not have any relevant building material information from the CoreLogic dataset used; therefore,
only Monroe County locations are included in this section's analysis. The listed exterior wall materials are aggregated into
three categories: "Concrete", "Wood", and "Other" (Fig. 5a). Habitability functions are then developed for the concrete and
wood categories as functions of maximum water depth, flow speed, and significant wave height (Fig. 5b-g). Habitability
functions are not generated for the other category since there is no similar defining feature within the group.

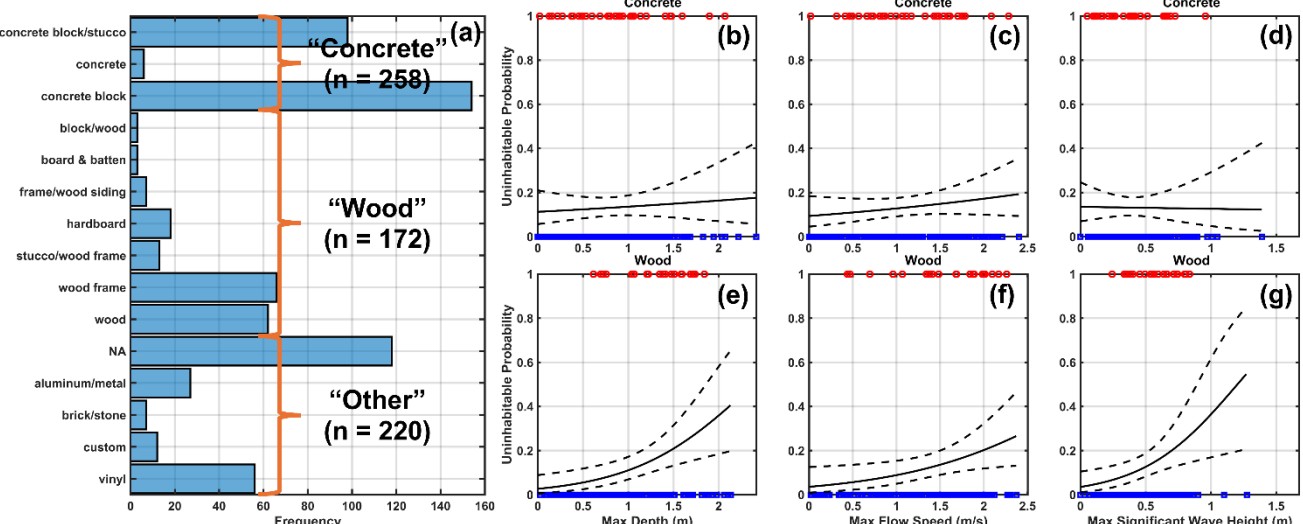

**Figure 5: Histogram of the exterior wall descriptions from CoreLogic for buildings analyzed in Monroe County and the three
aggregated categories: concrete, wood, and other (a). Building habitability as a function of maximum depth, flow speed, and
significant wave height for concrete (b-d) and wood (e-g). b-g use the same legend as Fig. 4.**

The only significant trends revealed from this analysis are for the habitability functions developed for the wood category (Table
3). The habitability functions developed for the concrete group are not significant at the 95% confidence interval. This can be
interpreted to mean that wooden buildings are less likely to be habitable after sustaining a relatively larger maximum depth,
flow speed, or significant wave height, while the uninhabitable probability of concrete structures is not influenced by the level
of hazard.





**Table 3: Coefficients for different building materials**

|  | Concrete | | | Wood | | |
|---|---|---|---|---|---|---|
|  | Depth | Flow Speed | Sig. Wave Height | Depth | Flow Speed | Sig. Wave Height |
| $\beta_0$ | -2.065** | -2.267** | -1.854** | -3.586** | -3.286** | -3.305** |
| $\beta_1$ | 0.217 | 0.349 | -0.080 | 1.504* | 0.958* | 2.751* |
| AIC | 204.809 | 203.814 | 205.108 | 121.550 | 126.063 | 123.855 |
| BIC | 211.915 | 210.920 | 212.214 | 127.845 | 132.358 | 130.150 |
| $\chi^2$ test p-value | 0.579 | 0.254 | 0.924 | 0.001 | 0.018 | 0.005 |

For individual coefficients: * p-value < 0.05, ** p-value < 0.001

### 3.3 Habitability functions based on additional hydrodynamic parameters

Habitability functions are also developed using the maximum unit discharge ($hv$), flow momentum flux ($\rho h v^2$), total water depth ($h + H_{sig}$), wave energy flux ($\frac{1}{16}\rho g H_{sig}^2 \sqrt{gh}$), and total force ($\frac{1}{16}\rho g H_{sig}^2 + \rho h v^2$) as the hazard level (Fig. 6), where $h$ is the water depth, $v$ is the flow speed, $\rho$ is the density of water (1,000 kg/m$^3$), $H_{sig}$ is the significant wave height, and $g$ is gravitational acceleration (9.81 m/s$^2$) . These additional hydrodynamic parameters have been shown to be significant drivers of flood damage in addition to the basic hazard parameters of depth, flow speed, and significant wave height (Diaz Loaiza et

al., 2022; Xu et al., 2023), motivating the following analysis on their influence of building habitability.

The additional habitability functions generated for maximum unit discharge, flow momentum flux, total water depth, wave energy flux, and total force all exhibit significant positive relationships with the probability of a building being uninhabitable (Table 4). Of these five parameters, the habitability function dependent on maximum wave energy flux has the worst fit with an AIC of 670.685 and BIC of 680.334. This is partially due to the outlying habitable building with a modeled maximum wave

energy flux of about 6,000 kW/m; however, even if this point is excluded the values of the AIC and BIC are still the worst at 668.044 and 677.690, respectively. Visually, the confidence interval is also the largest for the wave energy flux habitability function.

While the habitability function developed for maximum wave energy flux performs relatively poorly, the other functions developed based on the additional hydrodynamic parameters are comparable to those developed for depth and flow speed.

Habitability functions based on unit discharge, total depth, flow momentum flux, and total force all exhibit better fits than the functions generated based on flow speed (Tables 2 and 4). However, none of the habitability functions for the additional hydrodynamic parameters have a better fit than the depth-dependent habitability function.



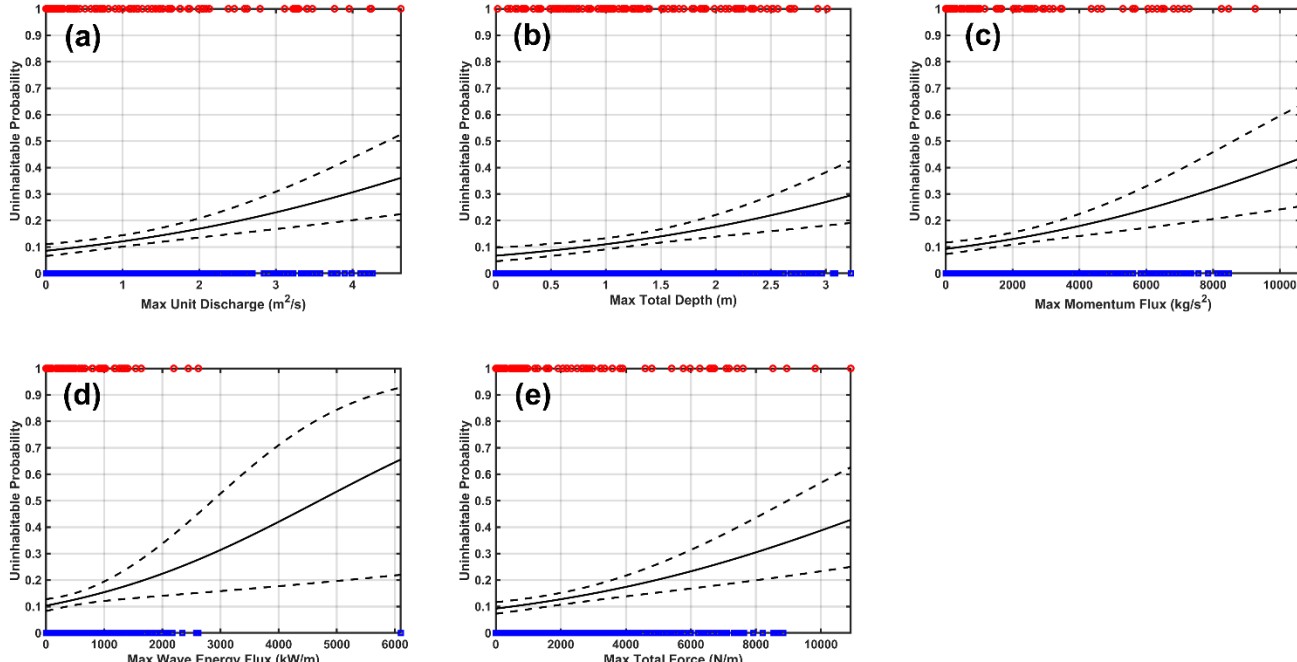

**Figure 6: Building habitability as a function of maximum unit discharge (a), total depth (b), flow momentum flux (c), wave energy flux (d), and total force (e). The legend is the same as in Fig. 4.**

**Table 4: Coefficients for maximum unit discharge, total depth, flow momentum flux, wave energy flux, and total force.**

|  | Unit Discharge | Total Depth | Momentum Flux | Wave Energy Flux | Total Force |
|---|---|---|---|---|---|
| $\beta_0$ | -2.370** | -2.635** | -2.283** | -2.164** | -2.289** |
| $\beta_1$ | 0.388** | 0.546** | 1.904e-04** | 4.604e-04* | 1.830e-04** |
| AIC | 661.171 | 662.527 | 661.780 | 670.685 | 661.919 |
| BIC | 670.820 | 672.176 | 671.429 | 680.334 | 671.567 |
| $\chi^2$ test p-value | 5.196e-05 | 1.064e-04 | 7.167e-05 | 0.009 | 7.712e-05 |

For individual coefficients: * p-value < 0.05, ** p-value < 0.001

Habitability functions for these additional hydrodynamic parameters are also developed for the concrete and wood building material categories described in the previous section (Figs. B1-B2). None of these habitability functions for concrete buildings are significant at the 95% confidence level (Table B1), but all those for wood buildings show significant positive relationships (Table B2). Furthermore, the wooden structure habitability function dependent on total depth ($h + H_{sig}$) has the greatest fit of all habitability functions developed for wooden buildings, including the one developed for just depth. Therefore, predicting



habitability can be improved by incorporating information on both inundation depths and significant wave heights at a building for wooden structures.

## 3.4 Habitability functions derived from multivariable logistic regression

Rather than combining the three basic parameters of depth, flow speed, and significant wave height into additional hydrodynamic parameters to develop habitability functions as in the previous section, multivariable logistic regression can be used as an alternative to derive habitability functions. This expands Eq. (2) into the following:

$$P(y = 1) = \frac{1}{1 + e^{-(\beta_0 + \beta_1 X_1 + \beta_2 X_2 + \dots + \beta_i X_i)}} \tag{3}$$

where the $i$ subscript indicates the $i$-th parameter in the regression model. Including multiple independent variables has been
shown to improve traditional depth dependent fragility functions (Charvet et al., 2015; De Risi et al., 2017), making it an important consideration for the habitability functions developed in this study. Four multivariable logistic regression models are considered (R1-R4), and Table 5 lists the hydrodynamic parameters considered for each model. The three basic parameters of maximum depth ($h$), flow speed ($v$), and significant wave height ($H_{sig}$) are considered for these models. To check for multicollinearity in these models, the variance inflation factor (VIF) is computed. All VIF values for these models are between
1.4 and 2.2, which is generally accepted as an indicator that multicollinearity problems are small (Sheather, 2009).

**Table 5: Hydrodynamic parameters considered for each multivariable logistic regression model.**

|       | R1 | R2        | R3        | R4        |
|-------|----|-----------|-----------|-----------|
| $X_1$ | $h$ | $h$       | $v$       | $h$       |
| $X_2$ | $v$ | $H_{sig}$ | $H_{sig}$ | $v$       |
| $X_3$ | -  | -         | -         | $H_{sig}$ |

Of the four multivariable models developed, R1 displays the best fit and R3 displays the worst fit (Table 6). While the AIC of R1 is very close to the AIC of the depth-dependent habitability function, the best forming univariable model, the BIC shows a greater preference for the depth-dependent function over R1. Furthermore, a likelihood ratio test to statistically determine if
R1 offers significant improvements over the nested depth-dependent habitability functions is performed. This likelihood ratio test accepts the null hypothesis, the nested depth-dependent function, over the alternative of R1 (p-value = 0.184). Therefore, it can be concluded that the habitability function developed that depends solely on maximum depth is the best estimator for predicting building habitability.





**Table 6: Coefficients for each multivariable logistic regression model.**

|  | R1 | R2 | R3 | R4 |
|---|---|---|---|---|
| $\beta_0$ | -2.760** | -2.686** | -2.464** | -2.776** |
| $\beta_1$ | 0.635* | 0.823** | 0.454* | 0.750* |
| $\beta_2$ | 0.230 | -0.060 | 0.124 | 0.352 |
| $\beta_3$ | - | - | - | -0.608 |
| AIC | 660.754 | 662.504 | 668.013 | 661.667 |
| BIC | 675.227 | 676.977 | 682.486 | 680.964 |
| $\chi^2$ test p-value | 8.302e-05 | 1.992e-04 | 0.003 | 1.798e-04 |

For individual coefficients: * p-value < 0.05, ** p-value < 0.001

Habitability functions based on the four multivariable models are also developed for the buildings in the concrete and wood categories (Table B3). However, none of the functions for concrete or wood structures based on these four models offer any serious improvement over those developed with the univariable models presented in Table 3.

**4 Discussion**

Overall, many of the habitability functions developed show that hydrodynamic hazard level significantly increases the probability of a building being uninhabitable following Hurricane Irma. This holds true for the first functions developed based on the three basic hazards of maximum flood depth, flow speed, and significant wave height, where the depth-dependent habitability function shows the best fit (Fig. 4 and Table 2). In an effort to improve upon this depth-dependent function, two methods for combining the basic hazard levels are explored. The first method creates new habitability functions based on five additional hydrodynamic parameters used previously to generate damage functions (Diaz Loaiza et al., 2022; Xu et al., 2023): maximum unit discharge, flow momentum flux, total water depth, wave energy flux, and total force. While the probability of uninhabitability has a significant positive dependency on these additional hydrodynamic parameters, none of these five new habitability functions have a better fit than the depth-dependent one (Table 4). This leads to the second method aimed at improving the habitability function dependent on depth, which is expanding the univariable regression to multivariable regression based on depth, flow speed, and significant wave height. The multivariable model R1 (depth and flow speed) shows a comparable AIC value to the solely depth-dependent function. This potentially aligns with previous studies that have shown including flow velocity in multivariable models improves fragility functions based on just depth (Charvet et al., 2015; De Risi et al., 2017). However, comparison of the BIC values shows a clearer preference for the univariable depth-dependent function.





This questions whether including maximum flow speed with depth in a multivariable model actually improves the ability to estimate building habitability. Results from the likelihood ratio test agree with those from comparing BIC values, suggesting the depth-dependent function is superior. Furthermore, using other hazard parameters besides depth increases the overall complexity of predicting habitability, which again points to the utility of the univariable depth-dependent habitability function.

This study also revealed significant differences in how varying hazard levels impact habitability probability for wooden and concrete buildings. None of the habitability functions developed for concrete buildings exhibit significant relationships between hazard level and uninhabitable probability. This indicates that other factors besides hydrodynamic hazards strongly influenced whether people returned to concrete structures after Irma. Conversely, the habitability functions developed for wooden structures display significant positive relationships between hazard level and uninhabitable probability, showing that hydrodynamic hazards strongly influenced if a wooden building became uninhabitable due to Hurricane Irma. These differences between wooden and concrete structures are understandable since flood hazards typically result in greater damage to wooden buildings than concrete ones (Charvet et al., 2015; De Risi et al., 2017; Suppasri et al., 2013).

While the habitability functions developed generally show the expected dependency of hazard level on building uninhabitable probability, there is still a good degree of uncertainty in estimating which buildings people return to. This is evident when visually inspecting the habitability functions, where some buildings are habitable at relatively high hazard levels and uninhabitable at lower hazard levels (Figs 4-6). This shows a major difference between traditional damage and fragility functions and these new habitability functions, where many socioeconomic factors can also influence if and when people return to a building after a flood event. For example, someone may not return to a completely undamaged building if they are able to stay with friends or family for an elongated period, and for others, returning to a highly damaged building may be the best option. While previous studies have looked at some of these factors influencing post-flood building habitability (Nofal et al., 2024; Paprotny et al., 2021; Paul et al., 2024; Yabe et al., 2020), this is the first study, to our knowledge, that directly quantifies how flood hazards influence habitability.

Besides uncertainties associated with socioeconomic factors, there are other assumptions and uncertainties in this study that could be addressed in the future. Uncertainty in the developed Hurricane Irma model is highly influenced by grid and DEM resolution, and higher resolutions are known to improve the flood model accuracy (Diaz Loaiza et al., 2022; Luppichini et al., 2019; Muñoz et al., 2024). The spatially varying Manning's roughness coefficients also introduce uncertainties in the flood model that influence the developed habitability functions. Aside from the flood model, the LBS data used to determine buildings that were uninhabitable due to Hurricane Irma bring their own uncertainties. For example, spatial inaccuracies of the LBS data could lead to misidentification of the associated building. Additional uncertainties could arise if the LBS data used is not representative of the study areas and populations (Swanson & Guikema, 2024). Finally, these habitability functions





could be improved if additional building information such as the number of stories or whether a building is elevated was available.

## 5 Conclusions

This study utilizes a Hurricane Irma flood model and LBS data to develop habitability functions for buildings in two Florida counties. First, we show that of the habitability functions dependent on maximum depth, flow speed, or significant wave height, the depth-dependent function performs the best. Five additional hydrodynamic parameters are also investigated to see if improvements can be made to the depth-dependent habitability function, but none of these additional parameters show increased performance. Then multivariable regression is employed, showing potential improvements to the univariable depth function with model R1 (depth and flow speed). However, additional analysis indicates these multivariable models do not offer significant improvements to the univariable depth-dependent function. Furthermore, buildings are grouped by material to evaluate how habitability functions compare for wooden and concrete structures, showing that the uninhabitable probability of concrete buildings is not influenced by hazard level while wooden buildings' uninhabitable probability increase with hazard level. These findings provide novel quantifications of the influence of flood hazards on whether a building becomes uninhabitable due to a flood event. This can be used in applications like HAZUS, which currently assumes buildings become uninhabitable for any nonzero flood depth (FEMA, 2024b). Future work could be done to incorporate socioeconomic factors into these habitability functions to increase the accuracy of estimating which buildings become uninhabitable during due to flooding.

**Data availability**

Elevation models are available from NOAA's National Centers for Environmental Information and the General Bathymetric Chart of the Oceans (GEBCO, 2023; NOAA NCEI, 2022). Land cover data comes from the 2019 National land Cover Database for the Contiguous United States (Dewitz & USGS, 2024). Meteorological data for Hurricane Irma is retrieved from the National Hurricane Center's revised Atlantic hurricane database and the Tropical Cyclone Extended Best Tract Dataset (Demuth et al., 2006; Landsea & Franklin, 2013). Tidal constituents are available from the Oregon State University Tidal Inversion Software (Egbert & Erofeeva, 2002). NOAA station data is available from NOAA's National Data Buoy Center (https://www.ndbc.noaa.gov/). The developed Hurricane Irma flood model can be shared upon reasonable request. Location Based Services (LBS) data, provided by Veraset, LLC, and CoreLogic property data are not publicly available.

**Author contributions**

BN, SG, and JB conceptualized the study. BN and JB developed the flood model and regression models. TS and SG analyzed the LBS data. BN drafted the manuscript and created the figures. All authors discussed and reviewed the final manuscript.



**Competing interests**

The authors declare that they have no competing interests.

**Financial support**

This project is funded, in part, with federal funds under award number NA23OAR4170115 from the US Coastal Research Program (USCRP) as administered by the US Army Corps of Engineers (USACE), Department of Defense, and the National Oceanic and Atmospheric Administration (NOAA) Sea Grant program, Department of Commerce. The content of the information provided in this publication does not necessarily reflect the position or the policy of the government, and no official endorsement should be inferred. The authors acknowledge the USACE, NOAA Sea Grant, and USCRP's support of

their effort to strengthen coastal academic programs and address coastal community needs in the United States. This work was also funded by a 2021 Catalyst Grants from the Michigan Institute for Computational Discovery and Engineering (MICDE).

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



**Appendix A**



**Figure A1: Maximum modeled flow speeds for Collier County (a) and the western (b) and eastern (c) regions of Monroe County. Building locations and associated maximum flow speeds used for habitability functions (d-f). To preserve privacy the exact building locations are not identified.**







**Figure A2: Maximum modeled significant wave heights for Collier County (a) and the western (b) and eastern (c) regions of Monroe County. Building locations and associated maximum significant wave heights used for habitability functions (d-f). To preserve privacy the exact building locations are not identified.**






## Appendix B

**Table B1: Logistic regression coefficients for maximum unit discharge, total depth, flow momentum flux, wave energy flux, and total force for buildings in the concrete category.**

|  | Unit Discharge | Total Depth | Momentum Flux | Wave Energy Flux | Total Force |
|---|---|---|---|---|---|
| $\beta_0$ | -2.050** | -2.014** | -2.013** | -1.911** | -2.011** |
| $\beta_1$ | 0.161 | 0.107 | 8.259e-05 | 6.385e-05 | 7.655e-05 |
| AIC | 204.438 | 204.978 | 204.405 | 205.089 | 204.463 |
| BIC | 211.544 | 212.084 | 211.511 | 212.195 | 211.569 |
| $\chi^2$ test p-value | 0.410 | 0.709 | 0.399 | 0.867 | 0.419 |

For individual coefficients: * p-value < 0.05, ** p-value < 0.001

**Table B2: Logistic regression coefficients for maximum unit discharge, total depth, flow momentum flux, wave energy flux, and total force for buildings in the wood category.**

|  | Unit Discharge | Total Depth | Momentum Flux | Wave Energy Flux | Total Force |
|---|---|---|---|---|---|
| $\beta_0$ | -2.978** | -3.716** | -2.675** | -2.632** | -2.708** |
| $\beta_1$ | 0.659* | 1.141* | 2.808e-04* | 0.001* | 2.776e-04* |
| AIC | 122.757 | 121.104 | 124.364 | 124.187 | 124.078 |
| BIC | 129.052 | 127.399 | 130.659 | 130.482 | 130.373 |
| $\chi^2$ test p-value | 0.003 | 0.001 | 0.007 | 0.006 | 0.006 |

For individual coefficients: * p-value < 0.05, ** p-value < 0.001





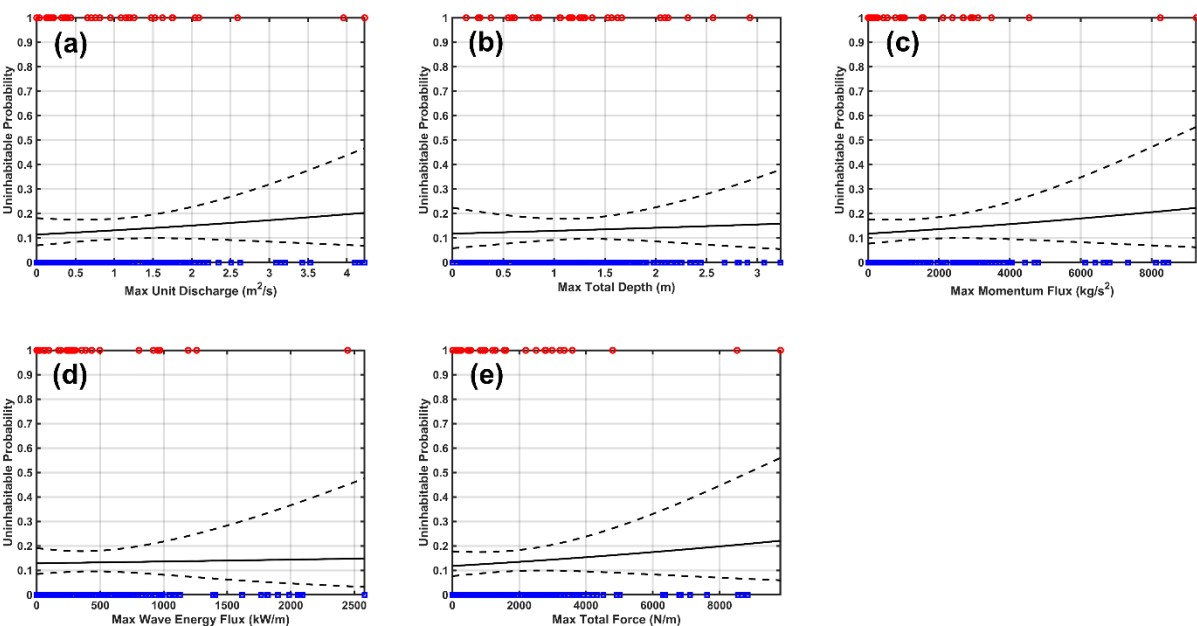

**Figure B1: Building habitability as a function of maximum unit discharge (a), total depth (b), flow momentum flux (c), wave energy flux (d), and total force (e) for buildings in the concrete category. The legend is the same as in Fig. 4.**

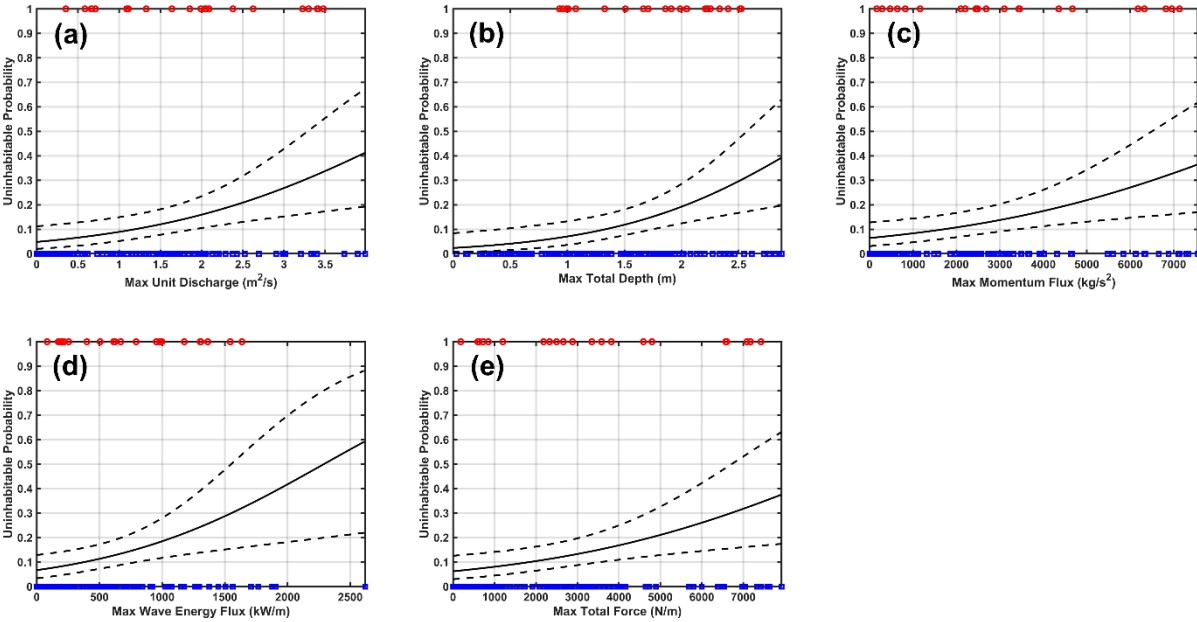

**Figure B2: Building habitability as a function of maximum unit discharge (a), total depth (b), flow momentum flux (c), wave energy flux (d), and total force (e) for buildings in the wood category. The legend is the same as in Fig. 4.**



**Table B3: Coefficients for each multivariable logistic regression model for buildings in the concrete or wood categories.**

|  | Concrete | | | | Wood | | | |
|---|---|---|---|---|---|---|---|---|
|  | R1 | R2 | R3 | R4 | R1 | R2 | R3 | R4 |
| $\beta_0$ | -2.254** | -1.959** | -2.118** | -2.150** | -3.848** | -3.753** | -3.832** | -3.955** |
| $\beta_1$ | -0.035 | 0.597 | 0.478 | 0.425 | 1.285* | 1.063 | 0.560 | 0.929 |
| $\beta_2$ | 0.363 | -1.065 | -0.736 | 0.426 | 0.360 | 1.309 | 2.229* | 0.301 |
| $\beta_3$ | - | - | - | -1.403 | - | - | - | 1.167 |
| AIC | 205.808 | 206.186 | 205.260 | 206.865 | 123.049 | 122.932 | 124.384 | 124.593 |
| BIC | 216.467 | 216.845 | 215.919 | 221.077 | 132.491 | 132.374 | 133.827 | 137.183 |
| $\chi^2$ test p-value | 0.520 | 0.628 | 0.395 | 0.522 | 0.005 | 0.005 | 0.010 | 0.011 |

For individual coefficients: * p-value < 0.05, ** p-value < 0.001