# Peer review of "Quantifying the influence of coastal flood hazards on building habitability following Hurricane Irma"

_EGUsphere, 2025_

## Author Comment (AC1)

**Response to Reviewer 1**

**This paper introduces a new method for quantifying storm hazards and their effect on building habitability, specifically in the context of Hurricane Irma. The proposed methods offer some insight into how hazard levels (e.g., water depth, flow velocity, etc.) can be used to estimate the probability of a building becoming uninhabitable following the hurricane. However, there are several missing discussions and insights.**

We sincerely appreciate the time that the reviewer has spent reviewing our paper and the comments and suggestions provided. We have revised our manuscript based on this review and responded to each comment below.

During our revisions, we have made some modifications/corrections to the flood model. Reviewer 1 asked about using an 80m SWAN grid and Reviewer 2 commented about increasing the model domain. These changes had minor impacts on our model and study. We also corrected two important details regarding the flood model during our revisions that resulted in more significant changes. Firstly, the original model used a uniform Manning's coefficient of roughness for the Monroe County portion of the model. After implementing a spatially varying Manning's coefficient, the maximum flow speeds modeled in Monroe County were reduced, which also resulted in better logistic regression fits for the models relying on flow speed. The second flood model detail we corrected is for the overland significant wave heights modeled in Collier County. We discovered a bug in the software that resulted in a lack of resulting wave data for Collier County. Our revised model now includes simulations that were not subject to this bug and therefore contain complete wave results for the full model domain.

**The criteria for building habitability are overly simplified and do not account for human behavior, such as voluntary displacement despite a building remaining structurally sound.**

We agree that our criteria for a building being considered habitable neglects factors like voluntary displacement. This is something we originally discussed in the Introduction in lines 45-47 and in the Discussion in lines 315-324. To further clarify our assumptions for building habitability, we have added to the Methods, "This assumes that the reason a user did not return to a location is solely because that location was damaged by Irma beyond habitability. This assumption does not account for other socioeconomic factors that may influence if and when someone returns to a location." In the Results, we have also added the sentence, "Another apparent detail of these functions is that some buildings are uninhabitable at relatively low hazard levels and others are habitable at relatively high hazard levels. This highlights some of the uncertainty in estimating building habitability using just hazard levels."

**The sample size is also limited—only 920 buildings were analyzed, with just 12% classified as uninhabitable—raising concerns about potential overfitting in the regression models.**

Since we are using relatively simple regression models for these habitability functions, our sample size is appropriate. For example, our must complex multivariable model, R4, includes 3 covariates, giving us 917 degrees of freedom.

**The model validation is less robust than expected; for example, it lacks overland flood comparisons and relies primarily on offshore water level gauges.**

Thank you for suggesting we add additional model validation. Following a new study by Asher & Luettich (2025), we have incorporated additional USGS storm tide sensor measurements that were temporarily installed during Irma. There are six of these USGS storm tide sensors located in our area of interest, but one of these sensors (FLCOL03171) is rejected during the quality assessment of Asher & Luettich (2025). Therefore, we now include five of these USGS storm tide sensors in our model validation. Based on these additional USGS sensors, our model performs comparably if not better than the validation shown in Asher & Luettich (2025). We have modified Figures 1 and 2, Table 1, and Section 2.2 based on this additional validation.

**Despite statistical significance, there is substantial overlap between habitable and uninhabitable buildings, which undermines confidence in hazard level as a strong predictor of uninhabitability.**

We concur with the reviewer's comment that there is overlap between the hazard levels experienced by habitable and uninhabitable buildings. To give greater confidence to using hazard levels to predict uninhabitability, we have included box plots that give greater detail on the hazard levels used for the habitability functions (Reviewer Figs. 1-3), showing differences in the distribution of hazard levels for habitable and uninhabitable buildings. We have included these box plots in the revised manuscript as Figs. A3, B3, and B4.

[Figure]

Reviewer Figure 1: Box plots of the maximum depth (a), flow speed (b), significant wave height (c), unit discharge (d), total depth (e), flow momentum flux (f), wave energy flux (g), and total force (h) used to develop habitability functions for Collier and Monroe counties.

[Figure]

Reviewer Figure 2: Box plots of the maximum depth (a), flow speed (b), significant wave height (c), unit discharge (d), total depth (e), flow momentum flux (f), wave energy flux (g), and total force (h) used to develop habitability functions for Monroe County buildings in the concrete category.

[Figure]

Reviewer Figure 3: Box plots of the maximum depth (a), flow speed (b), significant wave height (c), unit discharge (d), total depth (e), flow momentum flux (f), wave energy flux (g), and total force (h) used to develop habitability functions for Monroe County buildings in the wood category.

**The addition of a multivariable model does not improve predictive performance, calling into question the benefit of increased complexity.**

Our finding that the multivariable models do not improve performance does question the benefit of increased complexity, which we have argued points to the univariable models being superior for estimating habitability.

**Several improvements could strengthen this work: increase the sample size by including more regions, hurricanes, or buildings; revisit and expand the discussion of model limitations; clarify the assumptions behind using cell phone data as a proxy for habitability; and test whether altering the return-date threshold affects the results.**

We appreciate all the potential improvements suggested by the reviewer for strengthening this paper. We agree that increasing the sample size would improve this work. While we are limited to the data we have presented in this study, we have included discussion on the need for studying other regions and hurricanes to the conclusions: "Developing habitability curves for different regions and flood events is another area of future research that should be explored. Given this study focuses on two Florida counties, it would be insightful to investigate other regions both inside and outside the United States. Differences in building codes, zoning laws, and other policies may significantly change how flood hazards influence building habitability, which could be compared against the habitability functions developed here for Collier and Monroe Counties."

We have also expanded the discussion of the flood model limitations to include the uncertainty caused by the "parameterization of Hurricane Irma's wind and pressure field." For the

habitability models developed and using cell phone data as a proxy for habitability, we have added the additional discussion on limitations in the methods and results sections per our response to the earlier comment on the oversimplification of building habitability.

As the Reviewer suggests, we have tested whether altering the return-date threshold affects the results (Review Fig. 4). With the data utilized in this study, we only have access to earlier return-date thresholds, so we tested setting the threshold to returned by September 28th, 27th, 25th, and 20th, where the threshold used in the study are those that did not return by the 28th. As expected, this does change the habitability functions, shifting each habitability function towards greater uninhabitable probabilities. This is understandable since setting the return-date threshold to earlier dates is just changing habitable entries to uninhabitable. While we cannot test the opposite impact of making the return-date threshold later, it can be inferred that we would see the habitability functions shift towards lower uninhabitable probabilities.

[Figure]

Review Figure 4: Sensitivity of habitability curves for both Monroe and Collier Counties for five different return-date thresholds.

**Finally, the applicability of this method to regions outside the U.S. is unclear. Given the U.S.-centric dataset, it remains uncertain whether this approach could be generalized globally.**

The specific methods used for this study can be applied anywhere a flood model can be developed, Location Based Service (LBS) data is available, and property/building data is available. While the LBS dataset used in this study is U.S.-centric, Veraset LLC, the provider of the LBS data, has data from over one hundred countries. The property dataset utilized here is provided by CoreLogic, which also has data for other countries. These are not the only companies that provide LBS and property data, so our methodology can be applied to many regions outside the U.S.

**Line 24 - (or hurricanes) This phrase in parenthesis makes it seem that all tropical cyclones are hurricanes when hurricanes are tropical cyclones of a certain wind speed threshold, consider rephrasing.**

We agree including this phrase in parenthesis is misleading and have removed it.

**Lines 23 - 26. Sentence is confusing as written**

We have removed part of the original sentence "In the United States, tropical cyclones (or hurricanes) make up the majority of costs due to all billion-dollar natural hazards, resulting in almost 7 thousand deaths and over $1.4 trillion in costs (CPI-Adjusted) since 1980 (Smith, 2020)" to just read "In the United States, tropical cyclones have resulted in almost 7 thousand deaths and over $1.4 trillion in costs (CPI-Adjusted) since 1980 (Smith, 2020)."

**line 26 - 'The significant loss...' should be losses**

Thank you for the correction.

**line 28 - why are damage and fragility functions in quotes here but nowhere else?**

We originally included these in quotes since we were introducing the terms but have now removed the quotes.

**line 52 - household displacement 0 days after Irma implies that people evacuated? Wouldn't 1 day following the hurricane be a better metric for damage?**

We agree that household displacement 0 days after Irma implies evacuation and using 1 day following Irma could provide a better metric for damage. We have now added this consideration.

**lines 65-66 - Discussion of hurricane Irma should be in methods? Expand on estimated flood and storm surge volumes for the areas of interest.**

Thank you for the suggestions. We have now included additional details on the surge and flooding Irma produced in southwest Florida and the Florida Keys. The modified part of this paragraph now reads, "In Florida, water elevations reached 1.1 m and 1.7 m above mean sea level (MSL) at NOAA tide gages in Key West and Naples, respectively. Overland, the Florida Keys and southwestern Florida experienced maximum flood depths that exceeded 2 m (Cangialosi et al., 2021). In addition to storm surge, Irma caused widespread destruction from wind and wave hazards, displacing millions of people (Issa et al., 2018; Joyce et al., 2019)."

We have elected to keep this discussion of Irma in the introduction rather than the methods as it is providing background information of Irma and does not directly influence the methodology of the study.

**lines 69-70 - Does the cell phone data indicate that the buildings returned to their previous levels of occupation?**

Yes, the cell phone data indicates when users resume a "typical" pattern of visiting a home or work location that matches the visit pattern prior to Irma's landfall. These details are explained in depth in Section 2.3, so no changes were implemented for this introduction paragraph.

**line 83 - 'The extend the model is from...' Awkward phrasing**

This was a typo we have now corrected to "The extent of the model is from 12.94° N to 32.84° N and 98.01° W to 63.91° W, covering the majority of Florida (Fig. 1a)."

**line 85 - why was 80m chosen?**

We chose an 80m resolution for the D-Flow FM because this was in line with previous studies that modeled Irma, which listed their finest resolutions of 100-200m (Musinguzi et al., 2022; Li et al., 2021; Dobbelaere et al., 2022). The new study by Asher & Luettich (2025) does use a finer resolution of 20-60m resolution in nearshore areas, but as stated earlier, our validation is comparable if not better than the validation presented in here.

**line 87 - 150 m, why not overlap D-Flow and SWAN's resolutions?**

We originally used a 150m SWAN resolution due to memory limits of the computer running the model; however, we now have access to greater computing power and are able to use an 80m SWAN grid. This has been updated in the manuscript.

**line 96 - spell out HEC-RAS first**

Done.

**line 103 - explain what a spiderweb grid is**

We have rewritten this line to clarify that we are describing a spiderweb grid and added additional information. The revised version states, "Together, these datasets and the Holland model are used to develop a symmetric profile of Irma as a spiderweb grid. Spiderweb grids convey the atmospheric pressures, wind velocity magnitudes, and wind directions used in the flood model on a polar grid, where the origin of the grid represents the eye of the hurricane at each timestep (Deltares, 2022a)."

**line 109 - how was SWAN drag coefficient determined to be insufficient?**

During the model development, we found the default SWAN drag coefficient profile produced unreasonably small wave heights and periods. We have now included this reasoning for using an increased drag coefficient in SWAN.

**lines 109 - 115 - explain more clearly the methods on updating the drag profile and wind field values**

We have improved the clarity of this paragraph by first including why the original drag coefficient profile was insufficient, where the new sentence now reads, "It was determined that

the default SWAN drag coefficient profile, which relies on the Wu (1982) relationship, is insufficient for this modeling, producing unreasonably low wave heights and periods." The second part of this paragraph we have modified is regarding how the wind field values were updated to achieve this new drag profile. This part now states the following: "Due to the difficulty in prescribing a new drag profile in SWAN, implementing this increased drag profile was instead done by increasing the wind speed values by 25% in the spiderweb grids used by SWAN. This 25% increase to the wind speeds corresponds to the same wind wave growth due to the increased drag profile described by Eq. (1)."

**lines 174-176 - '...essential services such as power and schools were recovered by this point.' What point, when was this data obtained, what date and how long after landfall?**

The point referred to here is the end of September 28, 2017. To clarify this, we have reworded point to date. This was 18 days after landfall in Florida, which we have now included in the text. The LBS data covered August 1, 2017 until October 3, 2017, which we have also now included. We have updated the citation from Swanson (2023) to Swanson & Guikema (2024), which shows the range of school recovery following Irma of 9/20/2017-9/27/2017. We have also included an additional reference that shows power restoration beyond what is included in Hodge & Lee (2017), indicating that power is restored to almost all customers by September 28, 2017 (Mitsova et al., 2018).

**lines 178 - 183 - confusing paragraph, rework**

We have reworked this paragraph to the following: "Each location derived from the LBS data is then approximated to the nearest building by assigning it to the nearest CoreLogic coordinate, representing the center point of a property. This ensures each LBS datapoint corresponds to an actual building and provides information on the building material. In some instances, this results in multiple LBS datapoints being linked to the same building. For these buildings with multiple LBS datapoints, a building is assumed habitable if at least one LBS user returned to the building by the end of September 2017. A building is assumed uninhabitable if all corresponding LBS users did not return to the building by the end of September 2017. LBS datapoints farther than 0.001 decimal degrees from the nearest CoreLogic coordinate are excluded."

**lines 186 - 192 - this is methods**

This and the following paragraph have been moved to the previous section (2.3 Determining building habitability following Irma).

**lines 316 - 320 - This discussion is what I was waiting for the moment I first saw figure 4, can this be addressed earlier?**

We have elected to briefly address this earlier in Section 3.1 right after introducing Figure 4 as requested; however, we believe leaving this original paragraph in the Discussion section is the most logical as it is a general discussion of all the habitability functions developed.

**model validation - can the model be further validated with flow depths, not just offshore gauges?**

As we discuss above, we have further validated our model using five additional USGS storm tide sensors.

**Figure 4 - is this for all buildings in both counties? Please expand in figure caption, including the binary outcomes at the top and bottom axes makes this style of figure unconvincing of the habitability functions being useful tools, the fact that there are uninhabitable buildings at low or 0 depths, flow speeds or sig. wave heights make me think that the probabilities are suspect.**

This is for buildings in both counties, which we have now added to the figure caption. To show differences in the hazards levels for the binary outcomes, we have added Figures A3, B3, and B4 showing box plots of the hazard levels for the uninhabitable and habitable buildings.

**All figures in the style of figure 4 should have a legend, the captions should not make the reader turn back to figure 4 to remember what each lines means.**

We have added a legend to Figures 5, 6, B1, and B2 and removed the part of the captions referencing Figure 4's legend.

**Figure axes and ticklabels are very small, consider enlarging for the average user**

All axes and tick labels have now been enlarged.

**Figure 5, where is the confidence interval reasonable, because the dotted lines are only close for moderate values of the hazard included. - see notes on figure 4's binary outcomes**

Thank you for the concern about the reasonableness of our confidence intervals. For Figures 5 and B1 specifically, the confidence intervals for the concrete buildings aren't particularly important since these relationships are not significant. For all other habitability functions developed, the confidence intervals widen at larger hazard levels because we have fewer buildings that experienced these large hazard levels. We have included the following sentence in our discussion to address this: "the confidence intervals of the developed habitability functions typically widen at larger hazard levels due to a smaller number of buildings experiencing these large hazard levels, which could be improved by including areas that experienced greater flood impacts in future studies."

**Figure 6 - are we still only looking at the county that has building data or is this all data again?**

Figure 6 uses building data from both counties. We have added this to the figure caption and indicated in all figures and tables if just Monroe County or both counties are included.

References

Asher, T. G., and Luettich Jr., R. A.: A hindcast of coastal flooding from hurricane Irma, Ocean Model., 197, 102582, 10.1016/j.ocemod.2025.102582, 2025.

Dobbelaere, T., Curcic, M., Le Hénaff, M., and Hanert, E.: Impacts of Hurricane Irma (2017) on wave-induced ocean transport processes, Ocean Model., 171, 101947, https://doi.org/10.1016/j.ocemod.2022.101947, 2022.

Li, Y., Chen, Q., Kelly, D. M., and Zhang, K.: Hurricane Irma simulation at South Florida using the parallel CEST model, Front. Clim., 3, https://doi.org/10.3389/fclim.2021.609688, 2021.

Mitsova, D., Esnard, A-M., Sapat, A., & Lai, B. S.: Socioeconomic vulnerability and electric power restoration timelines in Florida: The case of Hurricane Irma, Nat. Hazards, 94(2), 689–709, https://doi.org/10.1007/s11069-018-3413-x, 2018.

Musinguzi, A., Reddy, L., and Akbar, M. K.: Evaluation of wave contributions in Hurricane Irma storm surge hindcast. Atmos., 13(3), 404. https://doi.org/10.3390/atmos13030404, 2022.

Swanson, T. and Guikema, S.: Using mobile phone data to evaluate access to essential services following natural hazards, Risk Anal., 44, 883–906, https://doi.org/10.1111/risa.14201, 2024.

---

## Author Comment (AC2)

**Response to Reviewer 2**

**The manuscript outlines a methodology for defining habitability functions, which are purported to be a more accurate reflection of the impact of natural hazards and the ability of a community to recover than earlier work on damage or fragility functions. The authors focus on the impact of Hurricane Irma on locations on the Atlantic coast of Florida. The method couples information from a hurricane surge model (Delft3D-FM) with information from location based services and property data to deduce if residents of damaged buildings have resumed normal routines, linking this deduction to habitability of their homes. They indicated that the impact of water depth (flood depth) appears to be the major influence on habitability, greater than wave height or water velocities.**

**I think this is very interesting, perceptive work. I do have a few comments:**

Thank you for the kind words regarding our work as well as the comments provided. We appreciate the time spent reviewing our study and have responded to each comment below, incorporating corresponding changes to our manuscript.

During our revisions, we have made some modifications/corrections to the flood model. Reviewer 1 asked about using an 80m SWAN grid and Reviewer 2 commented about increasing the model domain. These changes had minor impacts on our model and study. We also corrected two important details regarding the flood model during our revisions that resulted in more significant changes. Firstly, the original model used a uniform Manning's coefficient of roughness for the Monroe County portion of the model. After implementing a spatially varying Manning's coefficient, the maximum flow speeds modeled in Monroe County were reduced, which also resulted in better logistic regression fits for the models relying on flow speed. The second flood model detail we corrected is for the overland significant wave heights modeled in Collier County. We discovered a bug in the software that resulted in a lack of resulting wave data for Collier County. Our revised model now includes simulations that were not subject to this bug and therefore contain complete wave results for the full model domain.

**1) Figure 1 shows the model grid. This seems very small for hurricanes. The grid implies the assumption that water level changes generated outside the grid due to the hurricane are negligible. This may or may not be the case for this specific storm event, but is not generally the case, as many hurricane researchers using ADCIRC use their standard grid, which covers half of the Atlantic Ocean. There has been work that suggests that a small grid might miss surge forerunners and other possible motions that can cause additional damage aside from the main surge event. This might explain why the model is incapable of simulating the long surge buildup (Figure 2d). If the emphasis is on peak surge, then perhaps it doesn't matter, as the model seems to be sufficiently tuned to get the max surge right. But to what degree does this impact the velocities (and, in turn, impact the finding that the habitability functions developed with velocities perform poorly)?**

Thank you for the suggestions regarding the domain of our model. We have now expanded our domain to cover much more of the Gulf of Mexico and Atlantic Ocean (Reviewer Fig. 1). As stated above, this had only minor impacts to the model relative to other changes, but we have still revised our study using this expanded domain.

[Figure]

Reviewer Figure 1: Updated figure showing expanded domain.

**2) Also regarding modeling: the tidal conditions from Egbert and Erofeeva can be less accurate in very shallow water, such as that near the Bahamas, where the offshore boundary is located. Was this accounted for?**

We appreciate the additional comment regarding the model setup. We did not account for the influence of very shallow water on the accuracy of the tidal constituents. However, when we expanded our domain as in Reviewer Fig. 1, we did not see a noticeable change in the simulated tides, indicating the potentially less accurate boundary conditions in very shallow water did not significantly influence our results.

**3) While I understand the presumed connection between habitability and the resumption of a normal routine originating from the same dwelling, the definition of "habitability" might be somewhat ambiguous. After Katrina, many residents lived in their homes while being compelled to return to their routines, yet many of these homes had no power or water. These homes served as functional shelters but that shouldn't be confused with recovery, since they were far from recovered. In many cases, these residents were out of options. This might actually bias the reliability of these habitability functions against those with lower incomes and fewer options. I guess I would like to either see a clearer definition of "habitability" (i.e. dwelling with sufficient cover from the elements), or these habitability functions placed in a more general context.**

Thank you for the suggestion to clarify our exact meaning of building habitability. While we originally discussed some of the assumptions regarding our habitability functions in the Introduction in lines 45-47 and in the Discussion in lines 315-324, we agree that there is some ambiguity in our original definition of habitability that should be clarified. Firstly, we have added additional lines clarifying these assumptions by stating in the Methods that "This assumes that the reason a user did not return to a location is solely because that location was damaged by Irma beyond habitability. This assumption does not account for other socioeconomic factors that may influence if and when someone returns to a location." Furthermore, in the Results we have added the sentence, "Another apparent detail of these functions is that some buildings are uninhabitable at relatively low hazard levels and others are habitable at relatively high hazard levels." Regarding the specific example of Katrina, our methodology would need adjusting since we state that locations "are assumed to be uninhabitable due to damages caused by Irma since essential services such as power and schools were recovered by this point." This goes beyond assuming a building is just a functional shelter, which we believe is appropriate for this study. However, this would be an issue for applying our methodology for an event such as Katrina, where the Reviewer points out many residents returned to homes without power or water. We have added the following to address this assumption: "Another important assumption for our definition of building habitability is that essential services such as power and schools are recovered 18 days after Irma's landfall in Florida. While this assumption is appropriate for Irma (Hodge & Lee, 2017; Mitsova et al., 2018; Swanson & Guikema, 2024), flood events that cause longer recovery periods for essential services may create difficulties in estimating building habitability the same way." We also included the Reviewer's point about potential biases against people with fewer recovery options by adding that "For example, someone may not return to a completely undamaged building if they are able to stay with friends or family for an elongated period, and

for others, returning to a highly damaged building may be the best option, which may bias these functions against people with fewer recovery options."

**4) The method relies on the availability and amount of LBS data. Collier and Monroe Counties have almost half a million residents between them. What would be possible in a place like the Louisiana coast? St Mary Parish, the largest coastal parish, has barely 50,000 residents. Is there a lower limit on data which would make the method meaningless?**

This is a good observation and question. The amount of LBS data available for Collier and Monroe Counties is actually quite larger than the amount utilized to develop these habitability functions, largely due to the fact that we did not use locations with modeled maximum flood depths of zero. Specifically, in Collier County we have 16,769 identified locations but only use 348 for the habitability functions. In Monroe County, which had a population of approximately 77,000 in 2017, we have 1,736 locations and use 659 (~1% of the population) to develop these habitability functions. So even though there are almost half a million residents between Collier and Monroe Counties, we retained a significantly larger portion of locations in Monroe County. Therefore, in a place like St Mary Parish, one could estimate obtaining roughly 500 locations (1% of the population), assuming the majority of the area experienced flooding and a similar population to building ratio as in Monroe County. This would be sufficient for repeating our methodology. The company we obtained LBS data from, Veraset LLC, has also reported that it is increasing the quantity of LBS data collected, meaning that greater proportions of the population could be used for developing habitability functions for years after 2017.

---

## Author Response (AR2)

**Response to Reviewer 1**

**Thank you for addressing the concerns brought up by the reviewers of this paper. The main concern I have now is some minor font inconsistencies in the manuscript as viewed on a laptop screen.**

We thank the Reviewer for again taking the time to review our manuscript. To clarify potential font concerns, the font selection is as follows: Arial is used in all figures, Cambria Math is used for all equations and variables, and Times New Roman is used everywhere else. As some variables are listed throughout the main text, Cambria Math occasionally appears within Times New Roman text, but this is intentional. We did update the (1) and (2) after Eq. in the text from Times New Roman to Cambria Math to align with the font used for the equation numbers. Unfortunately, without specific font concerns mentioned, it is difficult to fully address this final reviewer comment. Additional changes include a few minor grammatical corrections.